# STRATEGIC RECOMMENDATIONS FOR IMPROVED OUTCOMES IN CONGESTION GAMES

## ABSTRACT

Traffic on roads, packets on the Internet, and electricity on power grids share a structure abstracted in congestion games, where self-interested behaviour can lead to socially sub-optimal results. External recommendations may seek to alleviate these issues, but recommenders must take into account the effect that their recommendations have on the system. In this paper, we investigate the effects that dynamic recommendations have on $Q$-learners as they repeatedly play congestion games. To do so, we propose a novel model of recommendation whereby a $Q$-learner receives a recommendation as a state. Thus, the recommender strategically picks states during learning, which we call the Learning Dynamic Manipulation Problem. We define the *manipulative potential* of these recommenders as the proportion of action profiles which can be induced in $Q$-learners and propose an algorithm for the Learning Dynamic Manipulation Problem designed to drive the actions of $Q$-learners to maximize the social welfare. We simulate our algorithm and show that it can drive the system to convergence at the social optimum of a well-known congestion game. Our results show theoretically and empirically that increasing the recommendation space can increase the manipulative potential of the recommender.

## 1 INTRODUCTION

Route RSs are a prominent example of RSs (RS) that promise to improve the traffic experience for all its users. In the simplest of cases, shortest path algorithms are applied directly to networking and traffic (Abolhasan et al., 2004; Zeng and Church, 2009) and extended to user-facing platforms like Google Maps and Waze (Luxen and Vetter, 2011; Wang et al., 2014; Dai et al., 2015). Consider however what happens if all drivers receive the same shortest path recommendation and follow it: the recommended path will congest and it may even become the longest by travel time. Drivers may even realise this inefficiency and learn not to follow recommendations. Therefore, leading route RSs must necessarily recommend routes that account for their effect on the system, and account for whether or not drivers will follow the recommendations. This interaction between drivers and recommendations is what we formalize in this paper.

To study the route recommendation problem, we propose an extension of the model of congestion games (Rosenthal, 1973), where we relax the rationality assumption by having the players as tabular $Q$-learning agents. We assume that the agents play repeated congestion games

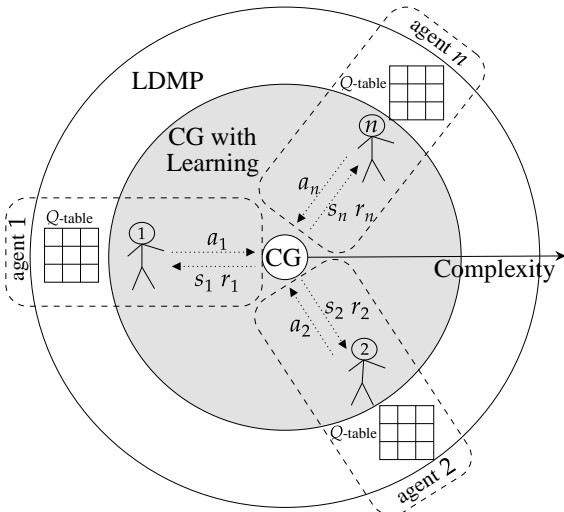

Figure 1: The state space of the Learning Dynamic Manipulation Problem (LDMP), where $Q$-learners are playing a congestion game (CG), and the $q$-tables, policies, and learning algorithms of the agents are known.

and learn over time which paths to pick, which attempts to provide a closer representation of real-life human behaviour.

We propose a model of recommendation whereby agents receive a recommendation at each iteration as a state and learn based on this state which road to pick. Thus, the problem becomes what we refer to as a Learning Dynamic Manipulation Problem (LDMP, see Figure 1), where the recommender's goal is to dynamically manipulate the learning dynamics of the agents. We frame the LDMP as a Markov Decision Process (MDP), given perfect knowledge of the $q$-values, policy, and update rule of the agents, which guarantees the problem to be computable in polynomial time (**P**-complete) with dynamic programming (Papadimitriou and Tsitsiklis, 1987). However, we show that the sizes of the state and action spaces are very large and argue that dynamic programming methods may be inadequate even in the best of cases (perfect knowledge of all agents). Therefore, we provide an algorithm that can compute user recommendations based on this information in reasonable time. We demonstrate that it can improve the social welfare of converged $Q$-learning agents in a classic routing game in which the Nash equilibrium social welfare is much worse than the socially optimal solution (Braess, 1968).

In the following sections, we describe the classical Congestion Game Model, following the literature. We then expand the model to incorporate learning agents, relaxing the rationality assumptions of the classical model. Then we formalize the LDMP and introduce a heuristic algorithm that can drive the system to better social outcomes. We present results that show the effectiveness of the algorithm. Moreover, we find that the *manipulative potential* of the recommender increases with the size of the recommendation space. We consider the potential of the algorithm and its limitations. Lastly, we discuss the relevance of our model with respect to real-life cases of route and other types of recommendation systems.

## 1.1 CONGESTION GAME MODEL

In congestion games (Rosenthal, 1973) $n$ players interact with a set of $k$ resources $\mathcal{A} = \{1, 2, \ldots, k\}$. Each player selects one resource $a$ (or more) and experiences a utility for that choice $u_a(f(a))$, where $f(a)$ is the number of players that picked resource $a$ and $u_a$ is a non-increasing function. This paper restricts attention to atomic, unweighted, symmetric, linear cost congestion games such that there are a finite number of players (atomic), each player contributes equally (unweighted), all players have the same available strategies (symmetric) and all utility functions $u_a$ are linear in the number of players that pick resource $a$ (linear cost).

The Nash equilibrium (NE) is an action profile $\boldsymbol{a} = (a_1, a_2, \ldots, a_n)$ in which no agents have unilateral incentives to deviate to a different action (Nash Jr, 1950). Congestion games are known to have a unique pure NE (Rosenthal, 1973) and are also a special case of convex potential games (Monderer and Shapley, 1996), games that admit a convex potential function whose maximization leads to the NE.

In a repeated congestion game, players will repeatedly play the game for up to infinitely many iterations. Here, selfish behaviour can sustain outcomes that are better than NE in terms of social welfare (Pareto-optimal) (Scarsini and Tomala, 2012), even though Best-Response dynamics are known to converge in congestion games. Therefore, it is interesting to look beyond Best-Response dynamics Candogan et al. (2013). Correlated equilibria are a well-known generalization of NE (Aumann, 1987), which contain equilibria that are possibly better than NE. A correlated equilibrium is achieved with a correlation device, meaning that all players are given additional information, which can change their best responses. A correlation device can be understood as a RS. For rational players and Best-Response dynamics in convex potential games, there are expected to be little possible improvements with correlation devices (Roughgarden, 2015; Koessler et al., 2023).

## 1.2 CONGESTION GAMES WITH LEARNING

This paper treats a bounded rationality case of repeated congestion games in which the players are $Q$-learning agents. As such, the congestion game can be framed as a Markov Decision Process (MDP), with the actions corresponding to the set of resources $\mathcal{A}$ and the reward function of the MDP being equal to the utilities experienced by the agents $R(\boldsymbol{a}) = (u_{a_1}(f(a_1)), \ldots, u_{a_n}(f(a_n)))$. The $Q$-learners are assumed to have an $\epsilon$-greedy policy, $\arg\max$ with probability $1-\epsilon$ and uniform random

with probability $\epsilon$, and update their $q$-values with the Bellman update rule. For the time-being the states have no explicit relevance to the congestion game, but will be explained in subsection 2.3 to represent recommendations. The $Q$-learners have their own state–action value function $Q_i$ : $\mathbb{S} \times \mathcal{A} \to \mathbb{R}$, a policy function $\pi_i : \mathbb{S} \to \mathcal{A}$, and an update rule $U_i : \mathbb{S} \times \mathcal{A} \times \mathbb{R} \times \mathbb{S} \to \mathbb{R}$ parametrized with learning rate $\alpha$ and discount factor $\gamma$. The environment runs for $\tau$ steps, and at each step $t$ a policy $\pi_i$ to determine their next action $a_{i,t}$, observe $(s_{i,t}, a_{i,t}, r_{i,t}, s'_{i,t})$ and then update $Q_{i,t}$ using their update rule $U_i$ to obtain $Q_{i,t+1}$.

Independent reinforcement learners that apply incremental updates to their policies are drawn towards the NE, but the NE may not be a stable equilibrium (Kleinberg et al., 2011; Bielawski et al., 2021; 2022). Additionally, the NE may not be the optimum of the social welfare function defined as $W = \frac{1}{n} \sum_i^n r_i$. It has already been shown, for some games (pricing game (Calvano et al., 2020; Klein, 2021), prisoner's dilemma (Schaefer, 2022; Dolgopolov, 2022)), that $Q$-learners are able to perform better social outcomes than the NE through implicit coordination. In Figure 2 we show that it is also the case for a Congestion Game with 100 players Carissimo (2023). This paper expands this result by focusing on the repeated case of congestion games played by $Q$-learning agents that will achieve even better social outcomes (than NE or the one reported in Figure 2) with the help of recommendations.

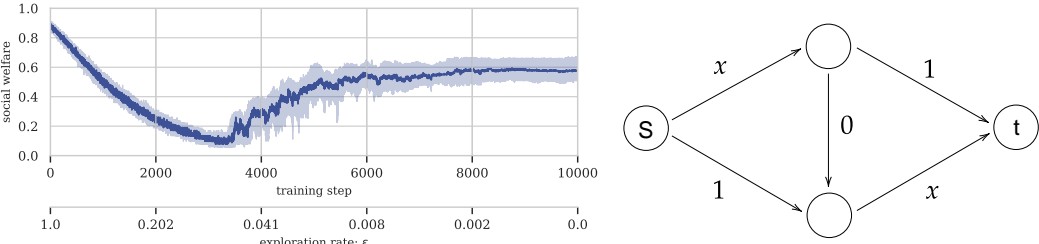

Figure 2: *Left:* Training curve of 100 $\epsilon$-greedy tabular $Q$-learners ($\alpha = 0.1$, $\gamma = 0.8$) in the Braess Paradox, which shows convergence to social welfare values much higher than the NE (0). *Right:* Illustration of the Braess Paradox network. Agents start in the "s" state and pick a path to reach state "t". The numbers represent the cost of traveling over a link. A cost of $x$ is the ratio of agents that choose that link. Rational and fully-informed agents all pick the crossing link in the augmented network (NE), which leads to high congestion and the worst possible social welfare.

### 1.3 FURTHER BACKGROUND

Route recommendation has also been tackled with collaborative filtering (Su et al., 2014; Cui et al., 2018), an effective tool from content recommendations (Su and Khoshgoftaar, 2009). RSs given strategic content providers have also been formalized in game theoretic terms (Ben-Porat and Tennenholtz, 2018). We also employ a formalism relying on game theory for our work. In the context of content recommendation there exist some fundamental limitations of incentive mechanisms, showing the challenges of designing a well-working recommender (Yao et al., 2023). Neural networks are also applied to handle large state spaces of personalised recommendations (Wang et al., 2019). Interestingly, the effects of route recommendations are simpler to estimate and theorize due to the similarity of traffic to congestion games (Monderer and Shapley, 1996) suitable for analysis with game-theoretic solution concepts.

As with content recommendation, route recommendation has also been identified to cause perverse congestion effects. Identified quickly in society, these trends are picked up by researchers showing that ubiquitous shortest path planning may worsen traffic congestion (Thai et al., 2016; Cabannes et al., 2017; Macfarlane, 2019), and possibly the economies of cities (Sweet, 2011). The dynamics such drivers create depend on road network topology, but most roads in cities were not designed with the assumption of widespread GPS routing recommendations. Congestion worsens when all users selfishly converge on the same shortest paths and may lead to a so-called "Price of Anarchy" (Koutsoupias and Papadimitriou, 1999), the ratio of the social welfare at the worst NE and the socially optimal solution.

In this paper, we propose a route recommendation algorithm for a well-known congestion game with a high price of anarchy called the Braess Paradox (Braess, 1968). The Braess Paradox demonstrates

theoretically that an increase in the capacity of a network (building new roads) can deteriorate its performance. This is because the added capacity may shift the NE of the network to an equilibrium that is less socially optimal. The Braess Paradox is found to be relevant for the routing of packets on the internet (Korilis et al., 1999; Tumer and Wolpert, 2000), the path selection of cars on roads (Argota Sánchez-Vaquerizo and Helbing, 2023) and the flow of energy on power grids (Schäfer et al., 2022).

Finally, the model of recommendation which we explain in the next section is novel but has some similarity to previous work on machine teaching (Zhu, 2015; Zhu et al., 2018). The goal of machine teaching is to select the optimal dataset that will allow a learner to learn the optimum. Recent work has extended machine teaching to reinforcement learning (Lewandowski et al., 2022). Our work is similar to machine teaching because we assume knowledge of a target distribution (like an optimum) to which we would like the system of $Q$-learning agents to converge on. It is also similar to machine teaching because the recommender in our model is capable of influencing the experiences of agents (picking data). Our model crucially differs from machine teaching because it involves picking the states for $Q$-learners rather than feeding them data (a state, action, reward, next state tuple for a reinforcement learner). Furthermore, our model has many $Q$-learners interacting in a congestion game such that what is optimal for an individual learner may not be socially optimal, leading to pluralistic notions of optimality.

## 2 LEARNING DYNAMIC MANIPULATION PROBLEM IN REPEATED CONGESTION GAMES

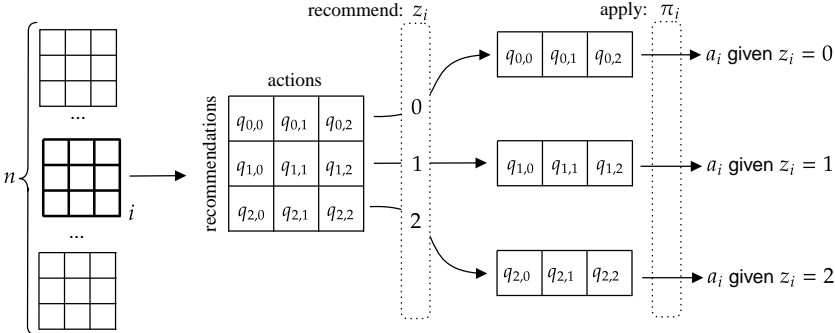

Figure 3: As each user learns a $q$-table of recommendation–action pairs, the recommender chooses the row of the $q$-table by recommending $z_i$ which determines the row to which the policy $\pi_i$ is applied. Time subscripts are omitted for clarity. This is a particular case where the number of actions equals the number of recommendations ($k = m$).

In this section, we discuss the main contribution of this paper which is a model of recommendations for $Q$-learners in congestion games. We assume that $Q$-learners are playing repeated congestion games and that a RS provides the $Q$-learners with recommendations. From here on out we will call the $Q$-learners which receive recommendations $Q$-users. $Q$-users receive recommendations as their states. Therefore, the MDP is extended to use states that reflect the recommendations. This could be understood as agents being given additional information. The $Q$-users learn the values of actions $a \in \mathcal{A}$ as a function of the recommendation states. To emphasize this particularity, we will denote recommendations by $z$, the recommendation space by $\mathcal{Z}$, and recommendation vectors by the bold $\mathbf{z}$. The RS outputs a length $n$ vector $\mathbf{z_t} = (z_{1,t}, \dots, z_{n,t})$ from a finite recommendation space $\boldsymbol{z}_{i,t} \in \mathcal{Z} = \{1, 2, ..., m\}$ and at each timestep provides each $Q$-user $i$ with a recommendation $z_{i,t}$. The $Q$-users receive this recommendation and select actions using their $\epsilon$-greedy policies $\pi_i(z_{i,t}) = a_{i,t}$. This process is visualized in Figure 3 for the case where $Q$-users have 3 actions ($k = |\mathcal{A}| = 3$) and the recommendation space has 3 elements ($m = |\mathcal{Z}| = 3$). The RS is effectively picking the rows of the $q$-tables that $Q$-users consider when applying their policy.

In subsection 2.1 we formulate the Learning DynamicManipulation Problem (LDMP) as finding the optimal policy of a Markov Decision Process. In subsection 2.2 we discuss how the LDMP scales as players ($n$), actions ($k$) or recommendations ($m$) grow. In subsection 2.3 we provide more theory

that can help understand the design principles of our algorithm, and justify the performance benefits that we report in 3.

## 2.1 REWARDS, STATES, AND ACTIONS

The policies of $Q$-users playing a repeated congestion game induce a distribution over actions $P_t(\mathbf{a})$ with domain $\mathcal{A}^n$ at each iteration $t$ where $\mathbf{a}$ is the action profile specifying an action for each agent. When the $Q$-users receive recommendations $\boldsymbol{z}_t$, the distribution depends on $\boldsymbol{z}_t$ which we denote as $P_{\boldsymbol{z}_t,t}(\mathbf{a})$. As all agents are treated to be equal (un-weighted congestion game) the congestion in the network is determined by mapping the distribution $P_{\boldsymbol{z}_t,t}(\mathbf{a})$ over action profiles to a simpler distribution over actions $P_{\boldsymbol{z}_t,t}(\mathrm{a})$ with domain $\mathcal{A}$ instead of $\mathcal{A}^n$. Then, we formulate the goal of the RS as trying to get as close as possible to the target distribution $P^*(\mathrm{a})$ which maximizes the social welfare of the congestion game. Define the cumulative discounted sum of future rewards as $G = \sum_{t=1}^{\tau} \gamma^t r_t$ for a given discounting factor $\gamma$. Then, the rewards $r_t$ are equal to the Kullback–Leibler divergence between the induced distribution and the target distribution:

$$r_t = -D_{\mathrm{KL}}\left(P_{\boldsymbol{z}_t,t}(\mathrm{a})\|P^*(\mathrm{a})\right). \tag{1}$$

We denote the reward function for the RS as $R_{RS}$. The goal of the RS is to find the policy $\pi^*_{RS}$ which outputs recommendations for the $Q$-users $\mathbf{z}$:

$$\pi^*_{RS} = \max_{\pi_{RS}} \mathbb{E}_{\sim\pi,\pi_{RS}}[\,G\,] = \max_{\pi_{RS}} \mathbb{E}\left[-\sum_{t=1}^{\tau}\gamma^t D_{\mathrm{KL}}\left(P_{\boldsymbol{z}_t,t}(\mathrm{a})\|P^*(\mathrm{a})\right)\right] \tag{2}$$

We assume the RS has a state space that includes the $q$-values of the users. For each agent, $Q : \mathcal{Z} \times \mathcal{A} \to \mathbb{R}$, and given $n$ agents, the space of all $q$-values is $\mathbb{S} = \mathbb{R}^{|\mathcal{Z}\times\mathcal{A}|n}$. Thus, the RS policy maps $\mathbb{S}$ to $\mathcal{Z}$, $\pi_{RS} : \mathbb{S} \to \mathcal{Z}$.

## 2.2 SCALING

The RS picks a recommendation $z_i \in \mathcal{Z}$ for each agent $i$. Given a recommendation space of size $m$ the action space of the RS becomes $\mathcal{Z}^n$ and thus has size $|\mathcal{Z}^n| = m^n$. In any practical application of $Q$-learning running on computers with finite precision, $\mathbb{S}$ will be finite. Given a finite action space, the system becomes an MDP defined by the tuple $(\mathbb{S}, \mathcal{A}_{RS}, T, R_{RS}, \gamma)$, where $T$ is the transition probability distribution $T(\boldsymbol{s}'|\boldsymbol{s}, \boldsymbol{z})$. For MDPs there are polynomial time algorithms to find the optimal policy. However, this MDP grows very fast. The state space $\mathbb{S}$ grows exponential in $n, m, k$: $\mathcal{O}(C^{nmk})$. The action space $\mathcal{A}_{RS}$ grows exponential in $n$ and polynomial in $m$: $\mathcal{O}(m^n)$. Nonetheless, the following theorems do not require the MDP assumption and give a sense of the extent to which a RS can influence the learning dynamics of systems of $Q$-users in congestion games.

## 2.3 DEMONSTRATING HOW RECOMMENDATIONS CAN MANIPULATE LEARNING DYNAMICS

We wish to demonstrate that state recommendations can *in theory* manipulate the learning dynamics. In so doing the RS acts as a controller of a non-linear discrete-time dynamical system. For non-linear discrete-time dynamical systems we can not guarantee *controllability* in a control theoretic sense Liu and Barabási (2016). Therefore we will resort to demonstrate some weaker notions than controllability, namely that under given assumptions it is possible to influence the actions that agents take. We then provide intuitions that can aid in understanding the design principles of our heuristic algorithm. In subsubsection 2.3.1 we prove that increasing the size of the recommendation space can increase the manipulative potential. In subsubsection 2.3.2 we show that it is possible to pick recommendations to optimize belief updates. In subsubsection 2.3.3 we show that it is necessary to vary recommendations in time to manipulate the system.

### 2.3.1 MANIPULATING BY INCREASING THE RECOMMENDATION SPACE

We will first present all of our arguments for the single agent case, which are then naturally extended to all agents to ensure that the entire dynamical system can be influenced. Proofs are in Appendix C.

Consider a single agent with an $m \times k$ q-table $\boldsymbol{Q}$ and a deterministic policy $\pi(z) = \arg\max_a \boldsymbol{Q}_{z,a}$.

**Definition 2.1.** (**Reachable**) We define the reachable set of an agent as the set of all actions which are an $\arg\max$ of a row $j$ of $\boldsymbol{Q}$.

$$\mathcal{R}(\boldsymbol{Q}) = \{a \in \mathcal{A} | \exists_z a = \pi(z)\}. \tag{3}$$

Our given $\boldsymbol{Q}$ is a q-table with $m$ rows which can be extended to have more rows. We define a new function $Ext$ which extends a matrix $\boldsymbol{Q}$ to a matrix $\boldsymbol{Q}'$ which has $m+1$ rows, and the first $m$ rows are identical to the rows of matrix $\boldsymbol{Q}$. $Ext$ is a function $Ext : (m \times k)$-matrices $\rightarrow (m+1 \times k)$-matrices so that $\boldsymbol{Q}' = Ext(\boldsymbol{Q})$ and $\forall$ indices $i, j$ where $1 \leq i \leq m, 1 \leq j \leq k$: $\boldsymbol{Q}'_{i,j} = \boldsymbol{Q}_{i,j}$. With the newly defined tools we can state our first theorem.

**Theorem 2.1.** (**Increasing Reachability**) When increasing the size of the recommendation space $m$ which amounts to adding rows to a q-table $\boldsymbol{Q}$, the size of the reachable set $R(\boldsymbol{Q})$ is monotonically non-decreasing.

$$\mathcal{R}(\boldsymbol{Q}) \subseteq \mathcal{R}(Ext(\boldsymbol{Q}))$$

The function $Ext$ can be seen as adding a new row which is a vector $\boldsymbol{v} = (q_1, ..., q_k)$ of q-values.

**Assumption 2.1.** (**Full Support**) The q-values of the new row $\boldsymbol{v}$ are drawn from a distribution $D$ where each action $a$ has a non-zero probability of being the $\arg\max$ of $\boldsymbol{v}$:

$$\boldsymbol{v} \sim D \text{ and } \forall a, P(\arg\max(v) = a) > 0.$$

**Theorem 2.2.** (**Global Reachability**) As the size of the recommendation space approaches infinity, and given Full Support of the q-values, the RS can induce any action in the $Q$-learner:

$$\lim_{m \to \infty} \mathcal{R}(Ext^m(\boldsymbol{Q})) = \mathcal{A}.$$

With these two theorems, **Increasing Reachability** and **Global Reachability**, we have shown that the size of the recommendation space has a big influence on the states that the RS can induce for a single $Q$-learner. If we have a system of $n$ $Q$-users playing a game, all of our results hold for the joint system too, with a tensor composed of all the q-tables of agents, $\mathbf{Q} = (\boldsymbol{Q}_1, ..., \boldsymbol{Q}_n)$, and the reachable set as the set $\mathcal{R}(\mathbf{Q}) \subseteq \mathcal{A}^n$ of action profiles $\boldsymbol{a}$. It will still hold that increasing the size of the recommendation space for any of the $Q$-users, and thus the number of rows in their q-tables, we can increase the size of the reachable set for the system.

**Definition 2.2.** (**Manipulative Potential**) Given a RS which can strategically pick recommendations $\boldsymbol{z}$, and users described by q-tables $\mathbf{Q}$ and $\arg\max$ policies, we define the manipulative potential of the RS as $\mathcal{R}(\mathbf{Q})/\mathcal{A}$.

The manipulative potential reflects the proportion of possible action profiles which a recommender can induce. If follows from the definition that maximizing reachability maximizes the manipulative potential. Fast forward to section 3, increasing the size of the recommendation set allows our algorithm to drive the system to near-system-optimum with finitely many states. In the next section we explain the other design principles required to achieve this performance.

### 2.3.2 MANIPULATING BY OPTIMIZING BELIEF UPDATES

Achieve higher social welfare requires *optimizing belief updates*. This is done to achieve a population of $Q$-users that believes the socially optimal actions to be better than the socially sub-optimal actions.

**Example** Consider a single user, and a single q-table

$$\boldsymbol{Q} = \begin{pmatrix} q_{1,1} & *q_{1,2} & q_{1,3} \\ q_{2,1} & *q_{2,2} & q_{2,3} \end{pmatrix} \begin{matrix} z_1 \\ z_2 \end{matrix} \quad \text{for actions } (a_1, a_2, a_3) \text{ and recommendations } (z_1, z_2).$$

We use an asterix $*$ to indicate the $\max$ values of the row. Both $q_{1,2}, q_{2,2}$ max values belong to the same column, $a_2$, so for both recommendations a $Q$-learner with an $\arg\max$ policy would pick

action 2. We know that the $Q$-learner's $q$-values change according to the Bellman update rule from time $t$ to time $t + 1$, and can express the total update as:

$$\Delta_{z,a} = q_{z,a}^{t+1} - q_{z,a}^t = \alpha \left[ r + \gamma \max_{a'} q_{z',a'}^t - q_{z,a}^t \right].$$ (4)

We also need the reward $r$ that the agent will observe so that we can predict the way the agent will update their $q$-value. In our presented model it is possible to predict the reward because the recommendations determine the rewards that agents will take. Then recommendations should be picked to optimize belief updates. Let us suppose that $a_2$ is a socially beneficial action. Given that it is the $\arg\max$ of both of the recommendation states, we can interpret this agent as having learned socially beneficial behaviour for both recommendations. If we wish to ensure that these socially beneficial actions will stay in the reachable set $\mathcal{R}(Q)$, we should use **positive reinforcement** and select $\arg\max_z \Delta_{z,2}$. If instead action 2 is socially sub-optimal, we should use **negative reinforcement** and select $\arg\min_z \Delta_{z,2}$.

### 2.3.3 CONSTANT RECOMMENDATIONS CHANGE NOTHING

If the RS picks a fixed recommendation vector **z** and never alters it during repeated play of the congestion game, the learning dynamics will resemble the case without recommendations, and collapse to the stateless congestion game as described in subsection 1.2. This is the case for any arbitrary recommendation vector, and means that static no-regret learning formalisms (Gordon et al., 2008) are not directly meaningful and applicable. To see this clearly, consider Figure 3 and imagine what happens when the same recommendation is picked for the entire horizon: the same $q$-values are used to select actions and update, which leads the system dynamics to be identical to the case where there are no recommendations. Therefore, the RS must vary the recommendations throughout the learning process to influence the learning dynamics, and it is not enough for the RS to find a recommendation vector **z** that induces the optimal action profile.

**Example** Take a new $\boldsymbol{Q} = \begin{pmatrix} *q_{1,1} & q_{1,2} & q_{1,3} \\ q_{2,1} & *q_{2,2} & q_{2,3} \end{pmatrix} \begin{matrix} z_1 \\ z_2 \end{matrix}$ $\begin{matrix} a_1 & a_2 & a_3 \end{matrix}$ , and assume $a_1$ is the socially beneficial

action. It is the case that $a_1 \in \mathcal{R}(\boldsymbol{Q})$, so we can induce action 1 by recommending $z_1$. However, after the $Q$-learner updates the $q$-table to $\boldsymbol{Q}'$, there is no guarantee that $a_1 \in \mathcal{R}(\boldsymbol{Q}')$. Specifically:

$$a_1 \notin \mathcal{R}(\boldsymbol{Q}') \text{ if } q_{1,1} + \Delta_{1,1} < \max\{q_{1,2}, q_{1,3}\} = \max\{b, c\},$$

and the update $\Delta_{1,1}$ will have altered the $\arg\max$.

## 3 RESULTS WITH BRAESS PARADOX

### 3.1 EXPERIMENTS[1]

In order to showcase the effectiveness of the proposed recommender algorithm we run it on a particular instance of a congestion game and compare the social welfare of the system over a fixed horizon. The particular congestion game setting is the augmented network of the Braess paradox (Figure 2), first introduced from the perspective of cars (Braess, 1968), but also extended for the cases of packet routing (Tumer and Wolpert, 2000). The NE has all agents crossing, which leads to an average latency of 2. However, the Social Welfare optimizing solution is for half of the agents to go up and the other half to go down, for an average latency of 1.5. The latencies experienced by the agents are linear in the fraction of agents that choose the actions, $\frac{n_u}{N}, \frac{n_d}{N}$. Specifically, the latencies, $l(a)$, are: $l(u) = 1 + \frac{n_u + n_c}{N}, l(d) = 1 + \frac{n_d + n_c}{N}, l(c) = \frac{n_u + n_c}{N} + \frac{n_d + n_c}{N}$. The rewards for agents are then the negative of the latency ($r_i = -l(a_i)$).

This scenario models a shared resource dilemma, which can lead to chaotic and non-convergent multi-agent learning dynamics. Figure S2 in the appendix gives additional information about the

---

[1]Code available at: `https://anonymous.4open.science/r/manipulateRL-57DB`

learning dynamics in the augmented network of the Braess paradox, which are not the focus of this paper. What is necessary to know is that: independent $Q$-users on this network are drawn towards the NE, and that the NE is unstable, and can cause oscillatory and chaotic dynamics. Therefore, it is particularly challenging for a RS to benefit social welfare by influencing the learning dynamics in this network.

We run the experiments for 9 population sizes between 100 and 900 agents. We also vary the number of states available to each agent (from 3 to 93). Three settings are compared with recommendations given by: our proposed heuristic algorithm, random, and no recommendations (*optimized*, *random*, and *none*, respectively). Each setting–population–state combination is run 40 times and the reported values are averaged over these runs.

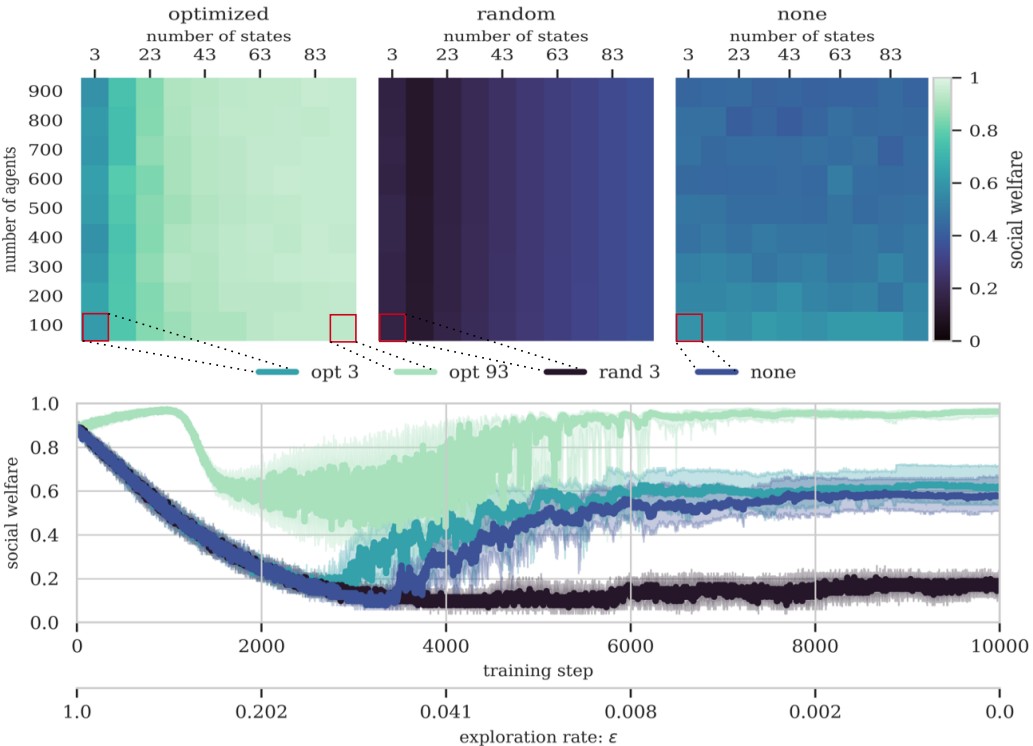

Figure 4: *Top:* Social welfare achieved in Braess's Paradox with varying numbers of agents and states for three recommenders: optimized, random, and none. *Botttom:* Evolution of the social welfare for four select conditions. Values of social welfare were rescaled from $[-2, -1.5] \rightarrow [0, 1]$, higher social welfare is better.

## 3.2 RESULTS

In Figure 4 we report the results of our experiments. The heatmaps presented at the top represent the average social welfare achieved over a finite horizon by each of the runs. We note that the heuristic recommender setting achieves consistently higher values of social welfare (than the *random* or *none* settings) regardless of the number of agents. The setting with no recommendations achieves better welfare than *random* recommendations which consistently give the worst results. Furthermore, we note that increasing the number of states for the recommender setting increases the social welfare of the system. At the bottom of Figure 4 we report the learning dynamics in terms of the social welfare (over the training steps and decreasing exploration rate $\epsilon$) for four conditions. The system with random recommendations with the least number of recommendation states ($m = 3$) converges to the NE. On the other hand, the system with no recommendations (*none*) converges to a better value than NE, but fails to reach the social optimum. In contrast, using the heuristic algorithm to recommend the optimized states shows some improvements. On one extreme end, for the system with the least recommendation states ($m = 3$), we reach social welfare that is comparable to *none*,

but with slightly faster convergence. But, on the other extreme of more recommendation states ($m = 93$), recommendations can steer the performance of the system to converge very close to the social optimum. It is worth noting that all systems exhibit a transient in the early phases of training (when the $\epsilon$ value is high).

## 4 Discussion and Conclusion

In this paper, we have formalized the Learning Dynamic Manipulation Problem (LDMP) which we propose as a model of route recommendations which take two main dynamics into account: the effects on the system, and the effects on the learning of users. We show that our model has a *manipulative potential* which increases with the size of the recommendation space. Furthermore, we propose a heuristic algorithm and demonstrated with a well known congestion game that we can influence the learning dynamics of a system $Q$-learners ($Q$-users). Figure 4 shows that an RS can improve the welfare of iterative games even when the NE is opposed to the social welfare optimizing solution. We interpret the success that any RS may have in such a system as due to its ability to i) slow down the dynamic evolution of the system which naturally "degrades" to the NE, and ii) speed up the dynamic evolution of the system, which, close to the NE, experiences a repelling force (Figure S2, right). Figure 4 also confirms our intuitions from Theorem 2.1 and Theorem 2.2 that the manipulative potential increases with the size of the recommendation space (number of states). This result could be interpreted as more information allowing for better social outcomes.

This model however allows for other interpretations. While in the presented example, the LDMP is directed at optimizing social welfare, nothing prevents the LDMP from being directed at other arbitrary metrics which may be harmful. Simultaneously, the model could also be seen as confusing the $Q$-learners which leads to a more grim interpretation that confusing the learning dynamics of users allows for greater manipulation. Such an interpretation may have interesting and relevant extrapolations to the present world where digital systems bombard their users with large amounts of information. For these reasons, we wish to emphasize that this formalism should be as alarming as it may be insightful. In subsection E.3 we include additional results and discussions about recommendation alignment and what it means for a route RS to be aligned with its users.

Our results have a notable limitation that we assume that the RS is omniscient, somewhat "God-like", such that the $q$-tables of all agents were known and the welfare optimizing actions for all agents were also known. This omniscience is a very restrictive assumption that will not apply in practice. Nonetheless, we wish to point out that while omniscience will likely not hold, RSs in practice all create models of user behaviour. While predictive power on an individual scale is harder, aggregating RS users into categories has been a successful means for RSs to model users. Therefore, even though it is unrealistic to assume omniscience, it is still interesting to consider what omniscience can afford while keeping in mind that omniscience is the best-case scenario of any RS. In subsection E.2 we run some simulations with relaxed assumptions of omniscience by adding noise to the $Q$-learning dynamics.

In future work on a theoretical side we are interested to better understand the controllability of multi-agent reinforcement learning systems both from an algorithmic lens, and as models of learning behaviour. We would like to extend our algorithm to larger congestion networks with Braess-like features Valiant and Roughgarden (2006); Chung and Young (2010). In this pursuit, we are interested to understand the proposed formalism in the context of other games. This sort of dynamic could also occur in other RSs, such as the centerpieces of the modern internet, that allow platforms to fetch and filter "optimal" results for their users (Ko et al., 2022). Such systems also create feedback loops with their users (never recommending certain content to a group of users, only showing users what they want to see, etc.), which can drive content bubbles and echo chambers. We propose that our approach may be used for RSs to optimize a notion of social welfare, while also achieving the necessary goals of a platform e.g. engagement. Both route RSs and content RSs need to account for the effects of their actions on the population they are affecting in order to consistently maintain high level of recommendations. Finally, while we believe it to be an intractable problem it would be very interesting to test the limits of solving the LDMP for congestion games by modelling the RS as a large Neural Network. We strongly encourage using heuristic methods to provide the bounds of comparison for such black-box techniques.

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

## A    Nomenclature and Symbol Glossary

| symbol | meaning |
|--------|---------|
| $n$ | number of agents |
| $k$ | number of actions |
| $m$ | number of recommendation states |
| $\mathbb{Z}$ | recommendation set |
| $z$ | recommendation |
| $\mathcal{A}$ | action set |
| $a$ | action |
| $\boldsymbol{a}$ | action profile (vector) |
| $\mathbf{Q}$ | $q$-tables of all agents |
| $Q$ | entry of $\mathbf{Q}$ |
| $Q_{i,z,a}$ | $q$-value for agent $i$ recommendation $z$ action $a$ |
| $Q$ | $q$-table of a single agent, a slice of $\mathbf{Q}$, $\mathbf{Q}_{i,:,:}$ |

Table 1: Nomenclature

## B    Learning Dynamics on the Braess Augmented Network

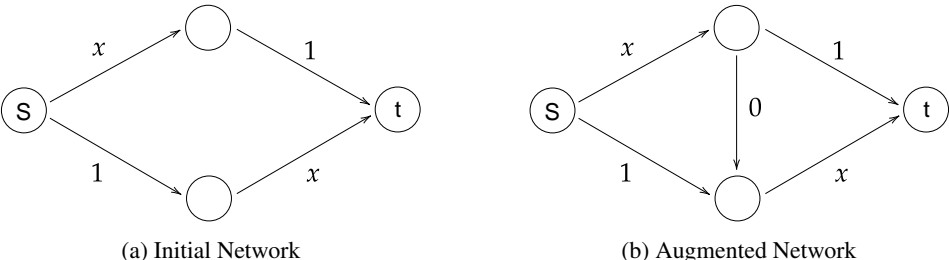

(a) Initial Network                    (b) Augmented Network

Figure S1: Illustration of the initial network (a), and the augmented network (b) in the Braess Paradox. Agents start in the "S" state and pick a path to reach state "t". The numbers represent the cost of traveling over a link. A cost of $x$ is the ratio of agents that choose that link. Two actions are possible in (a), $up$ takes the upper edges, and $down$ takes the lower edges. In (b) an additional action $cross$ is possible, which takes the first upper edge, crosses to the lower section at the middle, and finishes on the second lower edge. Rational and fully-informed agents all pick the crossing link in the augmented network (Nash equilibrium), which leads to high congestion and the worst possible social welfare.

## C    Heuristic Algorithm Pseudocode and Subroutines

### C.1    Proof of Theorem 2.1

*Proof.* Consider matrices $\boldsymbol{Q}$ and $\boldsymbol{Q}' = Ext(\boldsymbol{Q})$. We can split the $\boldsymbol{Q}'$ matrix into $\boldsymbol{Q}'_{1:m,1:j}$ and $\boldsymbol{Q}'_{m:m+1,1:j}$ (the row added in the $Ext$ operation). Since the $\mathcal{R}$ operates on rows independently and is associative we have $\mathcal{R}(\boldsymbol{Q}') = \mathcal{R}(\boldsymbol{Q}'_{1:m,1:j}) \cup \mathcal{R}(\boldsymbol{Q}'_{m:m+1,1:j})$. Since by definition of $Ext$ we have $\boldsymbol{Q} = \boldsymbol{Q}'_{1:m,1:j}$ we arrive at $\mathcal{R}(\boldsymbol{Q}') = \mathcal{R}(\boldsymbol{Q}) \cup \mathcal{R}(\boldsymbol{Q}'_{m:m+1,1:j})$. Therefore $\mathcal{R}(\boldsymbol{Q}) \subseteq \mathcal{R}(Ext(\boldsymbol{Q}))$. □

### C.2    Proof of Theorem 2.2

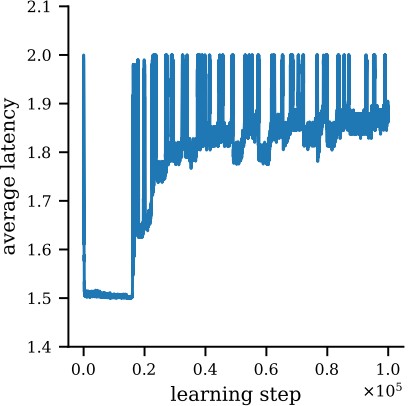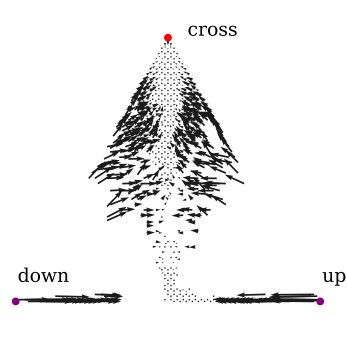

Figure S2: *Left:* The latency that emerges from the actions of the learning agents exhibits chaotic dynamics that tend towards the Nash equilibrium with $\mathbb{E}[l] = 2$. *Right* Vector field visualization of the evolution of the joint action space of the agents $(n_{up}, n_{down}, n_{cross})$. The corners of the triangle represent all agents taking a single action, and the arrows indicate the change between consecutive learning steps.

*Proof.* Take an arbitrary action $a \in 1, \ldots, k$ which has probability $p > 0$ of being the $\arg\max$ of every new added row. At any recommendation space size $m$ the probability that $a$ never appears as the $\arg\max$ of any row can be expressed as: $(1-p)^m$. As $m \to \infty$, the probability $(1-p)^m$ goes to 0. Therefore, with probability 1, the action $a$ will be the $\arg\max$ of at least 1 row, and the $a$ can be induced in an agent. □

## D   HEURISTIC ALGORITHM PSEUDOCODE AND SUBROUTINES

Algorithm 1 describes a routine that can be used to assign recommendations to $Q$-learners at each timestep of a repeated congestion game. In the algorithm, we call a few subroutines which we explain in greater detail.

The subroutine ESTIMATEREWARD calculates an optimistic estimate of the reward for each action $\bar{r}$. It calculates all the agents for which all recommendations only lead to one action. This defines a new congestion game with altered utilities $u'_a$ for each action, and a new socially optimal assignment $d'^*$. The reward for each action is estimated as $u_a(d'^*_a)$, the new socially optimal actions given the original utility for each action $u_a$.

The subroutine ESTIMATEUPDATE receives the estimated rewards per action and calculates the expected update as a difference $\Delta_{i,z,a}$ between the $q$-values of agents and the estimated reward assuming the $q$-values are updated with the Bellman equation.

The subroutine CALCULATEPRIORITY outputs a priority score for each action and reflects the actions that most need agents to be assigned to them. This is done by looking at the difference between the number of unique agents that could be assigned recommendations that lead to $a$, and the optimal value $d^*_a$. Priority 0 means that no more agents need to be assigned to the action.

The subroutine SELECTRECOMMENDATION receives a table with actions as columns and rows filled with the values $\Delta_{i,z,a}$ that have been sorted. The routine proceeds row by row, checking first if a column has priority 0, then the column is removed. Second, if the agent $i$ of $\Delta_{i,z,a}$ is unique in the row. If it is, recommendation $z$ will be assigned to agent $i$. If it is not unique then the recommendation is selected for the highest priority action. If the actions have equal priority then the recommendation $z$ is selected such that $\Delta_{i,z,a} < \Delta_{i,z',a'}$.

---

**Algorithm 1** Heuristic algorithm for a route recommender

---

**Require:** $Q$        ▷ the $q$-tables of all agents
**Require:** $n, k, m$        ▷ numbers of (agents, actions, recommendations)
**Require:** $d^*$        ▷ the target assignment of agents per action
 1: **for** $i \in \{1, 2, \dots, n\}$ **do**
 2:     $A_i \leftarrow \arg\max_a Q_{i,:,a}$
 3:     $P_i \leftarrow$ GETPOSSIBLEACTIONS$(A_i)$
 4: $r \leftarrow$ ESTIMATEREWARD$(P)$
 5: $\Delta \leftarrow$ ESTIMATEUPDATE$(Q, A, r)$      ▷ a table with update estimates separated by actions
 6: $C_{\mathbf{min}} \leftarrow$ SORTTABLE$(C, \min)$      ▷ sorts to minimize the update estimates by actions
 7: $\mu \leftarrow$ CALCULATEPRIORITY$(P, d^*)$      ▷ which actions need agents most
 8: $z \leftarrow$ SELECTRECOMMENDATION$(C_{min}, \mu)$      ▷ assign recommendations to achieve priority
 9: **if** $\nexists_{i \in agents}$ASSIGNED$(i)$ **then**      ▷ if agents are unassigned
10:     $C_{\mathbf{max}} \leftarrow$ SORTTABLE$(C, \max)$      ▷ sort to maximize the update estimates
11:     $\mu' \leftarrow$ CALCULATEPRIORITY$(P, d^*)$
12:     $z \leftarrow$ SELECTRECOMMENDATION$(C_{max}, \mu')$

---

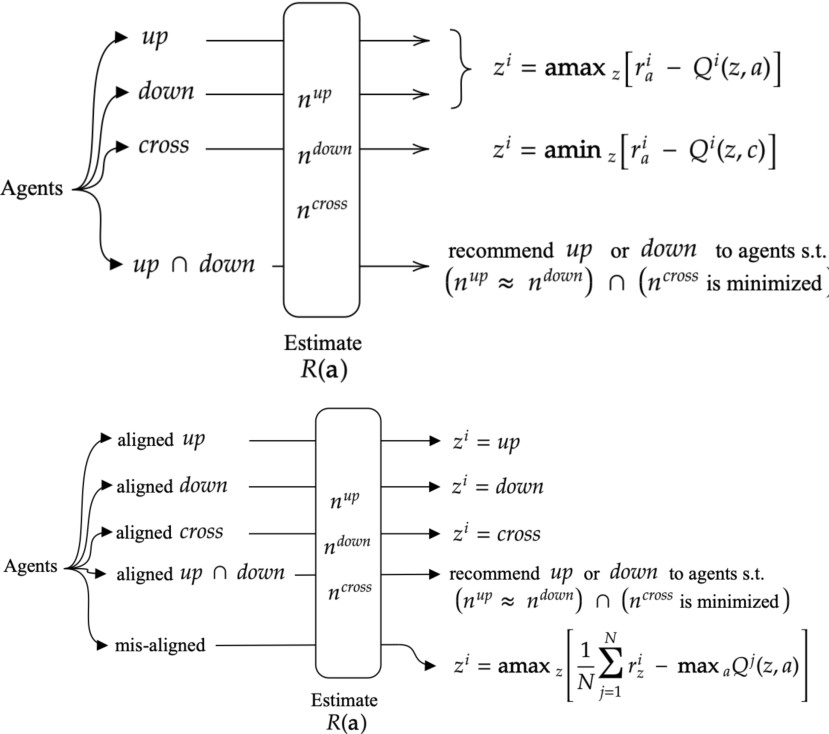

Figure S3: The Heuristic Recommender: Agents are categorized as $up$, $down$, or $cross$ groups according to the **argmax** actions in each of their recommendation states. In the case of the **argmax** being a tie between both $up$ and $down$, these agents are classified as $up \cap down$. Recommendations are then set such that the, $up$ and $down$ categories' $q$-values would improve most. Similarly, cross agents receive recommendations that worsen their $cross$ $q$-values most. Finally, $up \cap down$ agents are split equally between going up and down, if possible. The *Aligned* Heuristic Recommender: agents are categorized as $up$, $down$, or $cross$ groups according to **argmax** $= z_{i,t}$, that is that the recommendation $z_{i,t}$ is aligned with the **argmax** action. Agents can be classified into multiple groups. For agents that fall exclusively into a single group, they are recommended their aligned recommendation state. The agents that are both in the $up$ and $down$ groups are given (still aligned) recommendations such that they are split equally between going up and down, if possible. Agents with no aligned recommendation states are categorized as misaligned. All misaligned agents receive the same recommendation state, picked such that it is the state whose average belief change over all agents is maximized (the equation in the picture explains it best).

# E    THREE-STATE RECOMMENDER EXPERIMENTS

In the case where the route recommender can suggest $m = k$ recommendations to the agents, this is akin to the route recommender suggesting an action to the agent. For the initial ($k = 2$) and augmented ($k = 3$) Braess networks, we explored the potential of using a route recommender to manipulate the system of agents to achieve the social optimum (least latency). Figure S3 summarizes the heuristic route recommender for the augmented network.

## E.1    OMNISCIENT LEARNING DYNAMIC MANIPULATION

**When the RS is omniscient**    When $\bar{Q}_i(o, a) = Q_i(o, a), \forall i$, and the RS also knows $\pi_i \forall i$, the RS is said to be **omniscient**. In other words, the RS models the beliefs of agents perfectly, and with knowledge of the policies it can determine exactly what agents will do (a distribution over actions). Thus, when the RS is omniscient the problem is reduced to an MDP where the state information available to the RS fully determines the transition probabilities to future states conditioned on actions. In the omniscient case, reinforcement learning algorithms like $Q$-learning have polynomial worst-case guarantees $\mathcal{O}(n^c)$, even though the constant $c$ may be prohibitively large.

## E.2    NON OMNISCIENT LEARNING DYNAMIC MANIPULATION

**When the RS is not omniscient**    If the RS has an inaccurate model of beliefs, or it does not have knowledge of the policies of the users, the problem becomes a POMDP, where POMDPs are **NP**-hard in the worst case (Papadimitriou and Tsitsiklis, 1987; Mundhenk et al., 2000). The authors do not know if this particular problem admits an efficiently computable approximation (Lee et al., 2007), but hypothesize that it does not. For this reason, the examples in the following sections use heuristic recommendation algorithms instead of reinforcement learning. In the examples, it is assumed that: i) the users are $\epsilon$-greedy $Q$-learners ii) the RS is omniscient iii) the RS models users as greedy $Q$-learners. Therefore, when the $Q$-learners exploration rate $\epsilon = 0$, the RS is effectively omniscient, but when $\epsilon > 0$ the RS is no longer omniscient. Thus, $\epsilon$ will represent the likelihood that the RS makes an error in predicting the actions of users.

## E.3    RECOMMENDATION ALIGNMENT

**Recommendation Alignment**    We have thus set the stage to introduce what is meant by recommendation alignment. For a given agent $i$ at iteration $t$, we use the following definitions, assuming there are $|A|$ possible recommendations $z_{i,t}$ corresponding to each possible action, which users treat as state input to their $Q$-function. Given an observation $o \in \Omega$, a recommendation $z_{i,t}$ is *aligned* for user $i$ when $z_{i,t}$ is the action with the highest $q$-value in the state $(o \cdot z_{i,t}) \in \mathbb{S}$: $z_{i,t} = \mathbf{argmax}_{a'} Q_i((o \cdot z_{i,t}), a')$. Consequently, a recommendation $z_{i,t}$ is *misaligned* when $z_{i,t}$ is not the action with the highest $q$-value in the state $(o \cdot z_{i,t})$: $a \neq \mathbf{argmax}_{a'} Q_i((o \cdot z_{i,t}), a')$.

**Recommendation alignment can be a metric for trust**    Given that users are modelled as learners, an aligned recommendation suggests that users have learned to follow the recommendations being given by the RS. This can be taken as an indication that the user **trusts** the RS. Consequently, a user with a misaligned recommendation does not trust the RS for that particular recommendation.

**The Particular Case of Misaligned Agents**    In the case where an RS knowingly recommends a misaligned recommendation, we define the recommendation as a **manipulative** recommendation. Furthermore, it is possible for the beliefs of a user to have no aligned recommendations: for the users' $q$-values to lead him *not* to follow any of the recommendations. In such a case, it is not possible to provide the agent with a non-manipulative recommendation. However, it may still be possible to provide a recommendation that may establish alignment during learning.

**A Trade-off Between Welfare and Alignment**    The optimal policy $\pi_{RS}^*$ is guaranteed to maximize $G$, and at states $\mathbf{s}_t \in \mathcal{S}$ takes action $\mathbf{a}_t = (z_1, \ldots, z_N)$. The elements of $\mathbf{a}_t$ may or may not be aligned recommendations for all agents. Therefore, if a $RS$ were to restrict itself from providing misaligned recommendations, it may not be able to recommend the optimal recommendation. As

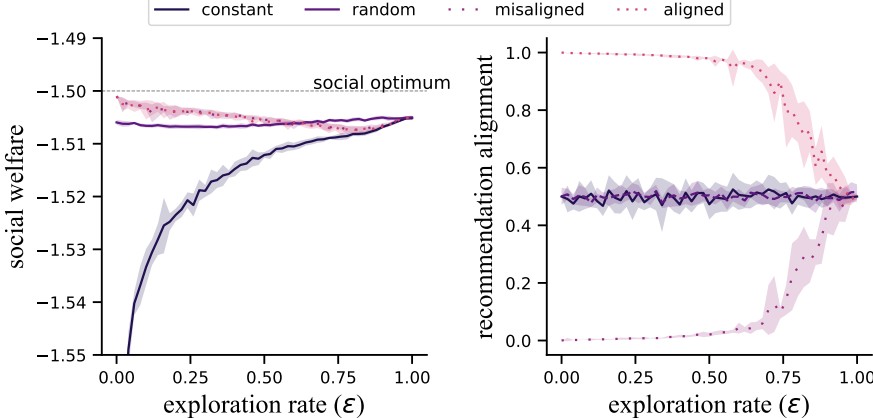

Figure S4: 100 $Q$-learners ($\alpha = 0.01$, $\gamma = 0$) are simulated for 500 learning steps in the initial network Figure S1a for varying RS and $\epsilon$ exploration rates. Curves plot averages 10 repetitions of training with error bars as standard deviation. *Left:* Social welfare for the different recommender schemes. With the exception of the constant recommender, recommendations can help the users achieve social optimum. *Right:* Alignment (and misalignment) for the heuristic recommenders converge to 0.5 as $\epsilon$ increases to 1 (fully random actions by agents.)

such, for some multi-agent systems, there can exist an inevitable trade-off between recommendation alignment and system welfare.

We have simultaneously conceptualized and demonstrated that the gains in social welfare could come at the loss of recommendation alignment. A mis-aligned recommendation means that a user has learned not to follow a recommendation, which we interpret as a metric of trust. The results of Figure S5 also lend credibility to the trade-off between welfare and alignment. The largest performance improvements were achieved by the **heuristic** RS which did not avoid misaligned recommendations and achieved poor alignment scores. On the other hand, the best alignment scores were achieved by the **aligned heuristic** RS which achieves lower optimizations of the welfare.

### E.4    Solving the Initial Network with Recommendations: A simple case for the LDMP

A symmetric routing game is considered with two routes: $N$ agents ($Q$-learners: $\alpha = 0.01$, $\gamma = 0$, $\epsilon = 0.01$) decide whether to pick up or down, leading to $n_u$ agents picking up, and $n_d$ agents picking down (Figure S1a). The latencies experienced by the agents are linear in the fraction of agents that choose the actions, $\frac{n_u}{N}$, $\frac{n_d}{N}$. Specifically, the latencies, $l(a)$, are: $l(u) = 1 + \frac{n_u}{N}$, $l(d) = 1 + \frac{n_d}{N}$.

The Nash equilibrium is aligned with the Social Welfare optimizing state, such that the average latency experienced by all agents is $1.5$. Using the negative of latency as the rewards ($r_i = -l(a_i)$), we choose to initialize all agents equally with the following $q$-values, $Q(\bullet, u) = Q(\bullet, d) = -1.5$. These are equivalent to the negative of the latency of the actions up and down experienced by agents at the Nash equilibrium (higher latency leads to lower $q$-values). Due to the $\epsilon$-greedy policy, for equal $q$-values an agent picks the first action in their $q$-tables (the definition of the **argmax** operator being in line with NumPy). Thus, regardless of the recommendation, the **argmax** of all agents will be to go up in the first iteration of the game. The only deviations from this action will be due to the $\epsilon$ exploration rate. This creates a simple scenario for an RS to evaluate the effects of its recommendations on welfare.

To demonstrate the potential of the RS, we test three cases in the Figure S1a scenario, report the results in Figure S4, and summarize them below.

**Tested cases and results for the Initial Network**    (**constant**) This recommendation is kept constant, and it does not help the agents to converge. In fact, keeping a recommendation constant is equivalent to providing no recommendation. The social welfare is poorest because, due to the

**argmax** definition, agents start all doing the same action, update their beliefs in the same way, and all do the other action in the next iteration. This is only offset by the $\epsilon$ exploration rate. (**random**) The recommendation is randomized between each iteration. The results show significant improvement from constant recommendation. While similar in social welfare to aligned and misaligned recommendations, only half of all recommendations are aligned. (**misaligned**) A two-step recommendation, sufficient for agents to immediately converge to the Nash equilibrium. The first recommendation is for all agents to go up. The second recommendation splits the population, as in the fixed case. The recommendation is then kept constant. While achieving rapid convergence to the Nash equilibrium this recommendation is fully misaligned. (**aligned**) A two-step recommendation, as in the misaligned case, but the first recommendation is for all agents to go down. Again, the second recommendation splits the population half-and-half and is then kept constant. This recommendation achieves both rapid convergence and recommendation alignment.

These results show that it is possible to **drastically improve $Q$-learning convergence** with cleverly timed recommendations that induce a coordinated behaviour in a system of multi-agent $Q$-learners. Additionally, recommendation alignment is measured as a metric that shows whether the coordination was achieved on average with aligned or misaligned recommendations. The difference between the aligned and misaligned cases is minimal but illustrates the effect that picking recommendations has on alignment.

### E.5 Initial $q$-values for Figure S5

The $q$-tables are initialized as the following matrix for each agent, aligned and misaligned respectively:

$$\begin{bmatrix} -1.5 & -2 & -2 \\ -2 & -1.5 & -2 \\ -2 & -2 & -1.5 \end{bmatrix} \text{ and } \begin{bmatrix} -2 & -1.5 & -2 \\ -2 & -2 & -1.5 \\ -1.5 & -2 & -2 \end{bmatrix} \tag{5}$$

### E.6 Solving the Augmented Network with the LDMP

**Testing different initial beliefs of users**   Figure S5 shows the relative performances for these RSs for two cases of initialized $q$-values: aligned $q$-values where the **argmax** for each recommendation state corresponds to the recommendation and misaligned $q$-values where the **argmax** does not correspond to the recommendation state (specific $q$-values are included in subsection E.5, results for two additional initializations of $q$-values are included in subsection E.7). It is noteworthy that in the misaligned initialization, each agent still has each action $up, down, cross$ as the **argmax** of one of its states, but it does not align with the recommendation state.

To compare and contrast different RS approaches we test the following RS which are visible in Figure S5.

**Tested cases and results for the Augmented Network**   (**constant**) The RS produces the constant recommendation $S = (u, d, u, d, \ldots, u, d)$. In practice, any constant recommendation is equivalent when all agents have their $q$-values initialized in the same manner. Furthermore, it is identical to the case where no recommendation is provided. This case is to demonstrate that a successful RS must actively, and dynamically change recommendation states for the agents to benefit the social welfare. (**random**) The RS generates a recommendation at each step that is uniformly picked at random from $S$. This case demonstrates that a random RS can have the beneficial effect of slowing down the natural learning dynamics. It also establishes a baseline recommendation alignment score. (**heuristic**) The RS attempts to pick recommendations such that as many agents as possible split between up and down, and the remaining agents are recommended actions that would lead their beliefs to change favorably due to the learning dynamics. A detailed picture of this heuristic RS is described in Figure S3. This RS achieves higher system welfare at the cost of recommendation alignment. The achieved performance increases the leverage of the manipulative potential of the omniscient RS system, which can be seen by poor recommendation alignment scores. The heuristic recommender results are nearly identical for average latency in the aligned (top row) and misaligned (bottom row) cases of Figure S5. This makes sense, as the heuristic recommender, "blind" to recommendation alignment, is not affected by the change in $q$-value initialization of the users. (***aligned***

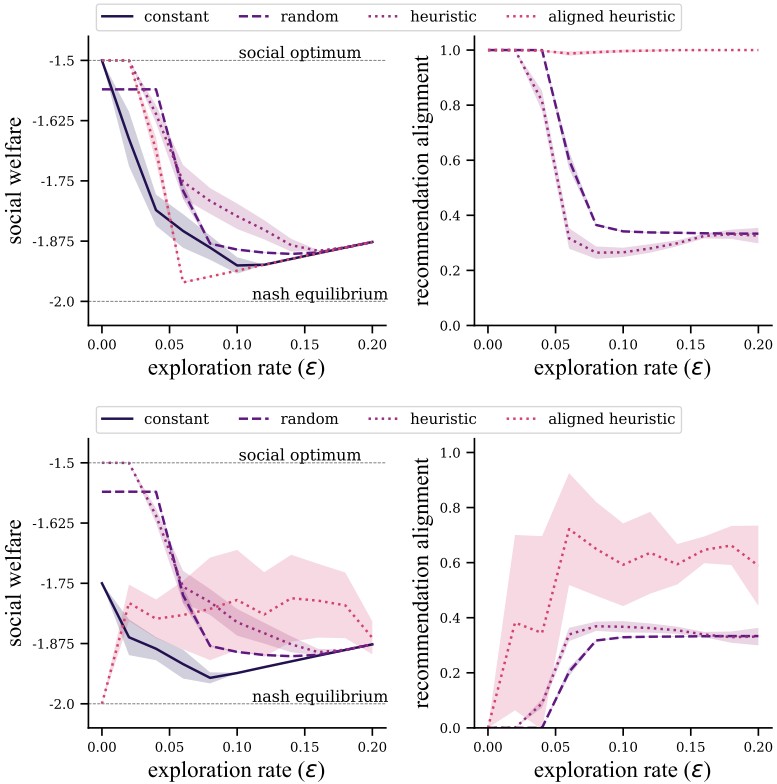

Figure S5: 100 $Q$-learners ($\alpha = 0.01$, $\gamma = 0$) are simulated for 10000 learning steps in the augmented network Figure S1b for varying RS and $\epsilon$ exploration rates. Curves plot averages 40 repetitions of training with error bars as standard deviation. Top row: The systems were initialized with aligned $q$-values (see Equation 5). The experiments show that the heuristic recommender yields the lowest latencies while using the aligned recommender results in high latencies (left column). However, the aligned recommender also maintains a high alignment with the beliefs of the agents (right column). Bottom row: Initializing misaligned $q$-values (see Equation 5) leads to a different evolution of the system, which also allows the aligned recommender to have lower latencies than all other recommenders at $\epsilon > 0.08$, while maintaining higher alignment.

**heuristic**) The RS attempts to pick recommendations such that as many agents as possible split between up and down, and the remaining agents are recommended actions that would lead their beliefs to change favorably due to the learning dynamics. A detailed picture of this *aligned* heuristic RS is described in Figure S3. This still achieves an improvement in the system welfare while prioritizing recommendation alignment and achieving the highest recommendation alignment scores. When possible, this recommender always recommends aligned recommendations.

### E.7 ADDITIONAL RESULTS FOR OTHER $q$-VALUE INITIALIZATIONS

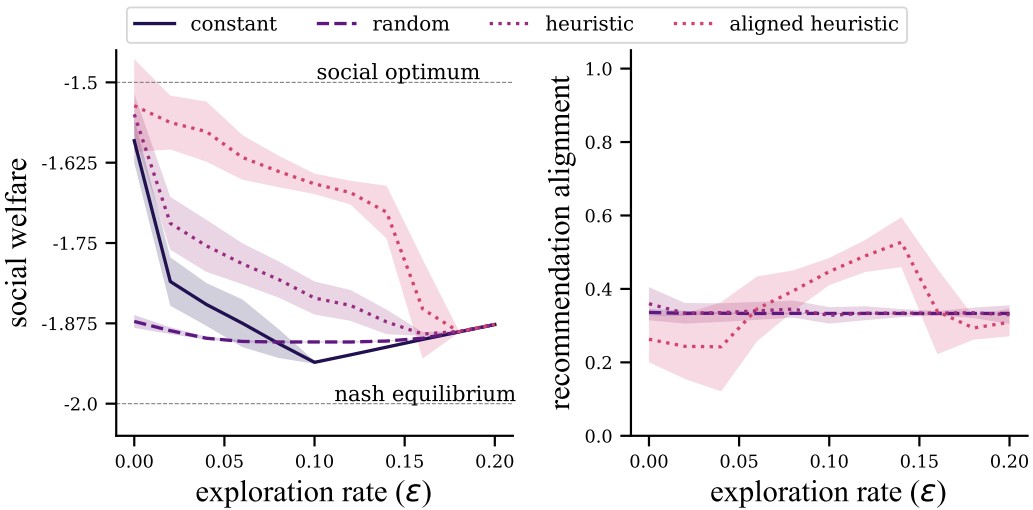

Figure S6: The systems were initialized to uniform random $q$-values.

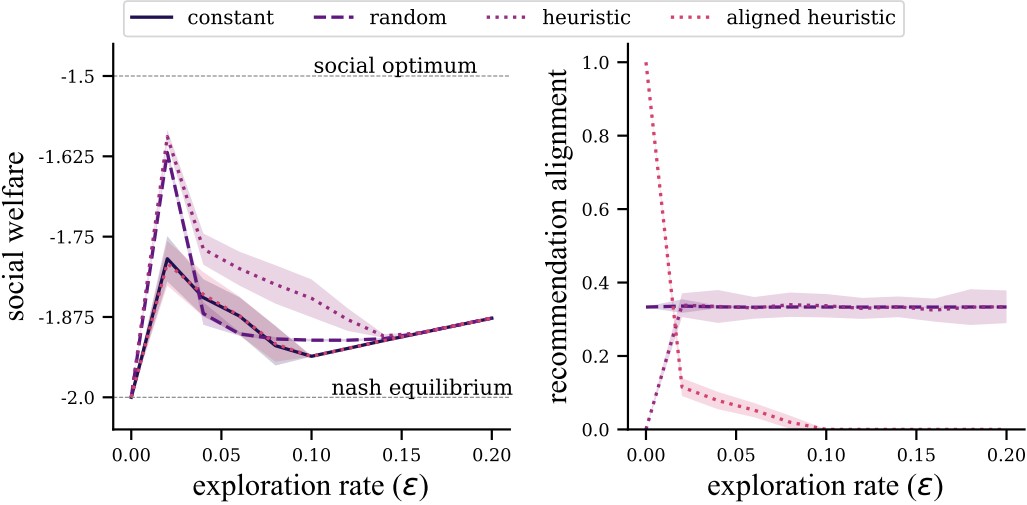

Figure S7: The systems were initialized to the Nash belief $Q = -2$.

