# OpenReview forum: "Strategic Recommendations for Improved Outcomes in Congestion Games"
_ICLR.cc/2024/Conference — Submitted to ICLR 2024_

### Official Review · Reviewer_rUVY · 2023-10-31

**Soundness:** 3 good
**Presentation:** 3 good
**Contribution:** 2 fair
**Rating:** 6
**Confidence:** 3

**Summary:**

This paper investigates the problem of performing recommendations for agents who repeatedly play congestion games to improve the social welfare. More specifically, this paper considers a situation where every agent is a Q-learner with states and he/she switches the state by the recommendation (Learning Dynamic Manipulation Problem, LDMP, in this paper). First, this paper defines the manipulative potential of a recommender and proves that it increases with the number of states, i.e., the possible kind of recommendations. Next, this paper proposes an algorithm for recommendation and empirically shows the effectiveness of this recommender by well-known Braess's paradox instance.

**Strengths:**

This paper investigates both theoretical and algorithmic aspects of LDMP. On the theoretical aspect, it proves that if more recommendation space is allowed, we can induce more combinations of actions that agents take. On the algorithmic aspect, it proposes a heuristic algorithm for recommenders that can improve the social welfare. Since this algorithm is not only intended to improve the social welfare, it can also be used for improving the other objective depending on the preference of recommender; which is of independent interest.

**Weaknesses:**

The main weakness of this paper is that the proposed heuristics is empirically evaluated on only a toy example of Braess's paradox that have only 3 actions. Although I admitted that toy examples like Braess's paradox is useful for investigating the behavior of the proposed and existing frameworks, the effectiveness of the proposed approach should be verified for more general and large datasets.
It is known that there are many instances that phenomenon like Braess's paradox occurs [1][2]. Therefore, the applicability for more larger congestion game should be empirically examined.

[1] Greg Valiant and Tim Roughgarden. Braess's paradox in large random graphs. In Proceedings of the 7th ACM conference on Electronic commerce, pp. 296--305, 2006. https://doi.org/10.1145/1134707.1134740

[2] Fan Chung and Stephen J. Young. Braess's paradox in large sparse graphs. In proceedings of the 6th International Workshop on Internet and Network Economics, pp. 194--208, 2010. https://doi.org/10.1007/978-3-642-17572-5_16

**Questions:**

In many congestion games including Braess's paradox, it is natural that the action of every agent is given as a combination of shared resources. In such a case, explicitly maintaining all the actions is prohibitive since the number of actions is generally exponentially large with respect to the number of shared resources. I think for some sets of actions (like path-sets of selfish routing setting), it may be possible to alleviate this issue because some computational components fall into easy problem like shortest-path problem; this is true for the equilibrium computation of congestion games. Do you have any idea for improving the time complexity of the proposed algorithm under such a situation where the set of actions is paths?

---

> ### Author Response · Authors · 2023-11-14
>
> Thank you very much for your review. With this first rebuttal we wish to answer your immediate questions. We are working on a reviewed manuscript that addresses your points that we expect to have ready in the next days.
>
> We wish to address the main weakness you point out in our paper which is that our heuristic model is only tested on the Braess Pardaox. We completely agree that this is not ideal, however we also believe that algorithms that work for other networks can be constructed to achieve similar results. What we believe our contribution to be is a demonstration of the **controllability** of the learning dynamics of q-learners which we them empirically verify in a congestion game.
>
> Our theorems were supposed to make this case, however we understand from the other reviewers that they need more work. We have found some relevant tools from the field of control theory that will help us make our claims stronger and mathematical, and will include these ASAP in our revised manuscript. We also wish to point out that our results contribute to a debate in game theory and congestion games on the effectiveness of recommendations. As mentioned in our paper, correlated equilibria which are the closest analogues of recommendations for rational players have been shown to be identical to nash equilibria in congestion games. This means that for rational players it is impossible to provide recommendations that would avoid an inefficient nash equilibrium. Our results therefore show that this is no longer the case for Q-learners, which is very interesting to consider given that Q-learning is a more realistic model of learning and decision making than the rational players assumed traditionally in game theory.
>
> We will be providing one more example of another network in the appendix which you may review. In this example our Q-learners face the initial network of the Braess Paradox where the Nash equilibrium is the same as the socially optimal solution. In this case, we show that with a cleverly designed recommendation "switch" it is possible to coordinate the Q-learners to rapidly converge on the Nash equilibrium.
>
> Finally, we wish to address your questions about scaling up the algorithm to large networks. This is clearly an interesting pursuit, and a very challenging problem. We believe that biggest hurdle in our assumptions is that we assume that the recommender knows all the q-tables of agents. Relaxing this assumption is necessary before we can scale up, but we believe that it may be possible. For example, the recommender may not know what the q-values of agents are exactly but instead can only infer it from the behaviour of the agents. Over time the recommender should be able to learn the q-values of the user. An even further relaxation may be to assume no knowledge of the q-values. We believe that it may even be possible to create a recommendation scheme that is able to achieve desirable results with no knowledge, but it will clearly have to be very different to a completely random scheme which have shown to be worse than no recommendations. One thing that is clear is that there must be correlations higher than random between the recommendations that different agents receive. If this last proposal were possible, then it would not even matter what the actions/paths available to the users are. The recommender system would operate with complete lack of this knowledge, and basically "bombard" all the users with correlated state information that influences their learning.

---

> > ### Comment · Reviewer_rUVY · 2023-11-23
> >
> > Thank you very much for detailed reply. I also partially read the updated manuscript, although the theoretical details were not carefully checked. I am still interested in scaling up the proposed framework to larger networks (with possibly exponentially many ), but I understand this is truly a future work.
> > I have also read the example of another network (you mean Appendix E.4?), but I think this example is still a toy example. I understand that these examples are indispensable to investigate the behavior of the proposed framework. However, I think a concrete examples of, for example, ten (or dozens) of actions, will be needed to verify the practicality of the proposed framework.

---

### Official Review · Reviewer_B8k7 · 2023-10-31

**Soundness:** 2 fair
**Presentation:** 1 poor
**Contribution:** 1 poor
**Rating:** 3
**Confidence:** 3

**Summary:**

The authors consider the atomic, unweighted, symmetric, linear cost congestion games and introduces a framework where an agent, playing the role of a recommender system (RS), learns to give recommendations to Q-learning agents (playing the role of players in the congestion games) in order to optimize social welfare. The authors then derive some results regarding the manipulative potential of the recommender. The authors then propose a heuristic algorithm of RS and show, by experimental means, that it can drive the system toward social optimum.

**Strengths:**

- The questions posed by the paper is quite interesting.
- The effort of the authors made in conducting a quite extensive experiment on the Braess 's paradox is applaudable.
- The presentation of the proposed framework is quite clear.

**Weaknesses:**

- In general, the writing is quite confusing. This includes a lack of robust mathematical presentations of results (Theorem 1 and 2) and several typos/grammar issues.
- The practicality aspect of this work is also questionable. As pointed out by the authors, the omniscience requirement for RS is a major limitation. Moreover, it is quite far-fetched to impose the Q-learning framework on all players in a congestion network.
- The main results (Theorem 1 and 2) do not give any guarantee on the convergence toward a social optimum (or at least, it lacks an explanation/justification). These theorems are not presented in a mathematically manner which hinders the readers to understand their contributions. As an heuristic scheme, the proposed algorithm is not supported by any performance analyses. It is quite disappointing as it can be seen from the experiments that as the number of states (recommendation options) increases, the proposed algorithm does drive the system closer to the social optimum.

- Some possible typos/ unclear sentences:
   + p.6, first paragraph, Regarding Theorem 1?
   + p.6, greater detail --> greater details
   + p.6 "....reflects the action that most need agents to be assigned to them"  ?
   + notation r is used both for welfare function W (page 2) and reward function (page 5)
   + In Equation (1), should z_t be a subscript instead of being a superscript?
   + In Section 2.1., there is a sentence saying that " the goal of RS is to find P*(a)" and another sentence saying that the goal of RS is to find the policy Pi^*_RS.

**Questions:**

- How the optimistic estimate $\bar{r}$ is computed in the heuristic algorithm ?
- What d*_a means exactly in the paragraph after Equation (3) ?
- Given a congestion game model, how an RS can design a recommendation set that guarantees Assumption 1?
- What is the relation between maximizing the manipulative potential and converging toward a social optimum?
- How precisely does the proposed algorithm confuses the Q-leaner (as remarked in Conclusion)?

**Details Of Ethics Concerns:**

No ethical issues with this paper to signify.

---

> ### Author Response · Authors · 2023-11-14
> **Preliminary rebuttal**
>
> Thank you very much for your review. With this first rebuttal we wish to answer your immediate questions. We are working on a reviewed manuscript that addresses your points that we expect to have ready in the next days.
>
> You and the other reviewers have pointed out that the theorems are to be significantly improved, and that it is not clear how the RS works from reading section 2. This clearly is a big shortcoming which we must address. We will get to that as soon as possible with the revised manuscript.
>
> **weaknesses**
>
> 1. We completely agree with you and the other reviewers on this point, and is a major part of our revision delivered to you ASAP.
> 2. Omniscience is a restrictive assumption which absolutely should be relaxed for future work. In the appendix we address this to an extent by showing that when the epsilon greedy users are kept at a constant epsilon (rather than decaying it) the RS is able to achieve an improvement for small values of epsilon but fails for larger values. This can be interpreted as the failure of the proposed heuristic algorithm to account for the actual exploration rate of the users.
> 3. Theorems 1 and 2 guarantee that given assumption 1 the system can be manipulated to be in any target state. That said, we completely agree with you and the other reviewrs that our formulation of the theorems is vague and must be reworked. As it stands we have found that there are two concepts of accessibility and controllability from control theory that should fit to phrase these theorems with mathematical rigor.
> 4. Thank you for pointing out all these typos. We will be sure to include them in our revision ASAP.
>
>
>
> **questions**
>
> 1. The optimistic estimate is computed by first categorizing all the agents according to the actions which could be induced by recommendations: the argmax actions of their q-tables. There will be some agents which have the same argmax for each recommendation, and others that have many different argmax values. We categorize them on the basis of their argmax values to determine the degree of flexibility that the system exhibits in that state. Then on this basis we create a new congestion game such that the agents that will only do one thing are treated as part of the congestion network. We then determine the socially best assignment of states and take the congestion in this scenario as the optimisic estimate.
>    We agree that this step is not well explained, so expect to include a pseudocode explanation in the appendix for the specific case of the braess paradox.
> 2. $d^*_a$  is the number of agents that take action $a$ at the social optimum.
> 3. Assumption 1 can not be guaranteed by designing the recommendation set, but is instead an assumption about the distribution of q-values, which the recommender system may not be in control of. We realized that assumption 1 must be reformulated and will do so ASAP in our review.
> 4. Maximizing the manipulative potential guarantees that you can induce any action distribution, including the socially optimal action distribution. We have realized that this is very similar to definitions of controllability from control theory, so we will be reformulating this theorem in these terms.
> 5. The proposed algorithm selects states for q-learners such that the actions selected by q-learners lead to desired outcomes. After each action the q-learners update their q-values for that recommendation. If there were no recommendations, the dynamics would settle on the behaviour shown in figure 2. We call this "confusion" without recommendations the Q-learners would act differently, and because the recommender is effectively disrupting the learning process of the Q-learners by switching around the recommendations.

---

> > ### Comment · Reviewer_B8k7 · 2023-11-22
> > **Responses to authors' feedback**
> >
> > Thanks for the responses. They address some, but not all, of my raised concerns. I would like to keep the original score.
> > (note that I cannot find, and hence, do not read, the "revised" paper mentioned by the authors)

---

> > > ### Author Response · Authors · 2023-11-22
> > >
> > > Dear Reviewer,
> > >
> > > Many thanks for your reply. We have re-uploaded the paper with the revised version. Are you sure that you can not see it? It works on our end. If you download the PDF it should be the new paper.
> > >
> > > Having checked out the new paper, would you be able to point out which concerns have not been addressed? We are happy to address any more of your concerns.
> > >
> > > The authors

---

### Official Review · Reviewer_fzBq · 2023-11-01

**Soundness:** 2 fair
**Presentation:** 2 fair
**Contribution:** 2 fair
**Rating:** 3
**Confidence:** 2

**Summary:**

This work considers the problem of designing recommendation systems to induce Q-learning agents in traffic networks to achieve near socially optimal outcomes; this problem is termed the Learning Dynamic Manipulation Problem (LDMP). In this model, agents are assumed to use greedy Q-learning to choose routes, and a central recommendation system can provide recommendations to each agent so as to influence the learning towards specific outcomes that minimize total congestion. This work proposes a heuristic algorithm for recommendation systems and shows that on the Braess paradox network, the heuristic algorithm can steer Q-learners towards more socially optimal outcomes, and this effect increases with the size of recommendation space.

**Strengths:**

The problem of inducing learning agents towards particular solutions is a very interesting direction on both the theoretical and applied side that is worth further explanation.

**Weaknesses:**

Certain parts of the modeling are not entirely clear and could benefit from more motivation or discussion (see below). The theoretical results (Theorem 1 and 2) appear somewhat unclear; they could benefit from more details or precision. On the experimental side, while the improvements for the improved recommendation system look quite interesting, they appear to be localized to a single network structure and it is not clear how general these results are.

**Questions:**

Recommendation: In full disclosure, my own interests are more on the theoretical side. Nonetheless, I think that this is a very interesting problem, but my current feeling is that this paper would greatly benefit from a clearer discussion of the modeling assumptions and choices. Since this is primarily an empirical paper, testing the heuristic algorithms on more general networks would also provide much better evidence for the utility of this approach. As a result, my initial belief would be to reject; that said, I could be convinced by the authors or the other reviewers to raise my score.

Comments/Questions

---Was $\varepsilon$-greedy Q-learning ever defined? My understanding was that this means that each agent uniformly explores with probability $\varepsilon$; if so, how does this connect with Assumption 1?

---What is the intuition behind the objective function of the RS in equation (2) in terms of KL divergence? Wouldn't a more natural objective be to minimize the weighted congestion of the actions of the Q-learners?

---Regarding Assumption 1: should there be a $t$ dependence somewhere here? If so, is that meant to hold for all $t$?

---It might be useful to provide pseudocode for the subroutines in Algorithm 1, at least in the Appendix.

---I had significant difficulty parsing the statement and content of Theorems 1 and 2. In Theorem 1, what does it mean that "the manipulative potential for every timestep $t$ is maximized"? The proof itself seems to use ``infinity'' in somewhat unclear ways that made it hard to understand. I also had issues determining a precise way to formalize the statement of Theorem 2. For both results, it might be very useful to give a more formal statement and proof to make clear what is meant.

---What is the (empirical or theoretical) computational cost of the heuristic algorithm, especially on generalizations to larger network topologies?

---My impression is that the empirical performance of the heuristic algorithm is quite interesting. But as mentioned above, my main concern is that these experiments seem to have only been done in the Braess paradox network and not on more general topologies.

---

> ### Author Response · Authors · 2023-11-14
> **Preliminary Rebuttal**
>
> Thank you very much for your review. With this first rebuttal we wish to answer your immediate questions. We are working on a reviewed manuscript that addresses your points that we expect to have ready in the next days. We are also interested in the theoretical side and hope that we can convince you that our results are an important initial step in understanding what may be provable.
>
> You and the other reviewers have pointed out that the theorems are to be significantly improved, and that it is not clear how the RS works from reading section 2. This clearly is a big shortcoming which we must address. We will get to that as soon as possible with the revised manuscript.
>
> To answer your direct **questions**:
>
> 1. epsilon greedy means, as suggested, that agents uniformly explore with probability epsilon. Your comment makes clear that we should define that explicitly. This is not connected with assumption 1 which is an assumption on the initial distribution of q-values. We realize however that our assumption needs to be rephrased to: assume that for all agents and for all actions, the q-values are drawn from a distribution such that each argmax has a non-zero probability of occurring. We will formalize this and get back to you ASAP.
>
> 2. If we understand correctly, you suggest that rather than looking at the KL divergence between action distributions and a target we minimize the congestion function directly. This point is well taken, and we believe that it will improve our presentation of the reward. In fact, what we care about is not the action distribution with a domain over $A^n$ but rather the action distribution with a domain over $A$: what is the proportion of players that selects each action? Thank you very much for pointing this out, we will modify it in the paper.
>
> 3. Assumption 1 should not have a t dependence. We agree that we must improve this theoretical section. We will do so and get back to you with well revised material ASAP for your review.
>
> 4. We fully agree that it would be useful to add pseudocode for the subroutines in the appendix, and see that it is not easy or intuitive to understand how the heuristic algorithm works from the way it is presented. We are adding these to the appendix.
>
> 5. We are going to fully re-work the theorems to be mathematical and get back to you ASAP.
>
> 6. The computational cost seems to be polynomial in the various sizes: the number of agents, the number of actions and the number of possible recommendations. We will try to give this full characterization to make it clearer.
>
> 7. Thank you for this point. We will do our best to provide some intuition in the revision of how the heuristic algorithm works. In the paper we provide a generalized presentation of the algorithm. Our results use a specific instantiation of if. Hopefully when we get back with our theorems it will be clear that it should be possible to design algorithms like our heuristic one for different kinds of networks. We have another example of a network which we will include which is the initial network of the Braess Paradox. In this network the Nash Equilibrium is the socially Optimal Solution, and we show that we can rapidly increase the speed of convergence to the equilibrium by cleverly picking some recommendations. We will include this in the revision ASAP.

---

### Official Review · Reviewer_rQFP · 2023-11-02

**Soundness:** 2 fair
**Presentation:** 1 poor
**Contribution:** 2 fair
**Rating:** 5
**Confidence:** 3

**Summary:**

The authors proposed a novel model of recommendation in which a set of Q-learners receive recommendations as their states and play a congestion game repeatedly. An heuristic algorithm was proposed to drive the actions of Q-learners toward a target action distribution for optimizing social welfare. Experiments show that the proposed algorithm can drive the system to convergence at the social optimum of a specific congestion game.

**Strengths:**

The proposed problem setting is timely and interesting and the high-level idea of recommending states to influence agents' beliefs sounds reasonable.

**Weaknesses:**

The writing of section 2 needs to be significantly improved. I'm not able to understand how the RS works from reading section 2.1. For example, what is the meaning of the space Z? And intuitively why we should believe the heuristic algorithm can achieve what we want? The description of Algorithm 1 is directly followed by the experiment results, and useful discussions are missing. Theorem 1 and 2 are not stated in rigorous ways. Could you quantify what does it mean by 'slower' and 'smaller'?

The assumption that the RS has access to the q-values of the Q-learners might be overly strong. In reality, it is unreasonable to believe that the platform has full understanding about each agent's belief state. It would be more interesting to relax this assumption.

The experiment is only performed on Braess Paradox instances, which seems limited and less convincing to me. It would be nice to demonstrate a comprehensive result on various congestion game instances.

**Questions:**

1. For the motivation, could you justify why it is reasonable to study the Q-learning agents in routing games, given that it is known that any better-response dynamics converges to the NE of congestion games.

2. the description in section 1.2 is confusing to me: it is said that the congestion game can be framed as a stateless MDP, but the formulation of Q-learners involves the state S. What is the state S here? And is a player allowed to observe other players' actions or rewards? It is beneficial to add a detailed description of players' Q-learning policy.

3. could you give an intuitive explanation why Q-learning agents can escape from the low-welfare NE of the Braess Paradox game even without the help of recommendations? I wish I could get some takeaway insight in this section but unfortunately the discussion is too little.

4. In terms of the better social outcome result, it is known that the for smooth games (including the class of congestion games), the PoA under NE is essentially the same as the PoA under CE/CCE [1]. So I suppose in general, the social welfare improvement from steering the NE to CE in congestion games can be very little. Do you agree with this argument?

5. there are some related works, e.g., [2,3] regarding mechanism designs for content recommendation platform, which also involves dynamical optimization over a congestion game structure. It would be nice to add some discussion in the intro or related work.


[1] Roughgarden, T. (2015). Intrinsic robustness of the price of anarchy. Journal of the ACM (JACM), 62(5), 1-42.

[2] Yao, F. et al. (2023). Rethinking Incentives in Recommender Systems: Are Monotone Rewards Always Beneficial?. Advances in Neural Information Processing Systems, 37.

[3] Ben-Porat, O., & Tennenholtz, M. (2018). A game-theoretic approach to recommendation systems with strategic content providers. Advances in Neural Information Processing Systems, 31.

---

> ### Author Response · Authors · 2023-11-14
> **Preliminary Rebuttal**
>
> Thank you very much for your review. With this first rebuttal we wish to answer your immediate questions. We are working on a reviewed manuscript that addresses your points that we expect to have ready in the next days.
>
> You and the other reviewers have pointed out that the theorems are to be significantly improved, and that it is not clear how the RS works from reading section 2. This clearly is a big shortcoming which we must address. We will get to that as soon as possible with the revised manuscript.
>
> To answer your direct **questions**:
>
> 1. Better-response dynamics converge to Nash, yes, but these dynamics are non-constructive. What do we mean. We mean that they do not provide the best response, they just assume that the best-response is played and thus Nash equilibrium is an almost trivial equilibrium point of these dynamics. Q-learning on the other hand is a constructive algorithm which can provably converge in MDPs. Its widespread application justifies its study in this model. Furthermore, as is shown in section1, Q-learning does not converge to the Nash equilibrium, which further justifies why we can not rely on best response analysis to know what to expect from practical RL algorithms when deployed IRL.
> 2. Section 1.2 was introducing repeated q learning in congestion games without recommendations, and stateless. Thank you for your c0mment, we agree that it is confusing to present it with the state space S and propose that we remove it in the revised version. Section 2 introduces the recommendations, so the q learners aquire a state as a recommendation.
> 3. The intuition is that the Nash equilibrium is unstable, since any multi-player deviation from the Nash will result in better outcomes for the deviating players. Q learners are able to implicitly coordinate, as is shown in other literature on prisoners dilemma, pricing games and the Braess Paradox. These are very interesting papers which show that Q-learning in some cases can lead to implicit coordination.
>    1. Calvano, Emilio, et al. "Artificial intelligence, algorithmic pricing, and collusion." *American Economic Review* 110.10 (2020): 3267-3297.
>    2. Dolgopolov, Arthur. "Reinforcement Learning in a Prisoner's Dilemma." *Available at SSRN 4240842* (2022).
>    3. Carissimo, Cesare. "The Social Benefit of Exploration in Braess’ S Paradox." *Available at SSRN 4348118* (2023).
> 4. This is a very interesting point. Indeed the PoA under CE and CCE are the same. This is because there are no CE/CCE which are different from Nash Equilibrium, also one of the results from Koessler et. al 2023 we cite in our paper. Thank you for the additional reference for such result. What this means for rational players is that there are no recommendations that can reduce the Price of Anarchy. This is quite the strong result. Our paper, shows that there are recommendations that can reduce the Price of Anarchy when the players are Q-learners. Given that Q-learning is a more realistic and constructive algorithm, we try to conclude that understanding whether or not recommendations can influence the outcome IRL can not be determined from rational player assumptions. Indeed the rational player assumptions would say that recommendations can not help avoid inefficient Nash equilibria.
> 5. Thank you for the added references that we were not aware of. We will look at them closer and see how they may be added to the discussion. [2] very interesting, as they think about designing rewards such that content creators are effectively playing a potential game. [3] takes again the perspective of content providers (or creators) which may strategically create content that will be recommended by a mediator (recommender system)
>
> Thank you for all comments. These are our introductory responses. In the next days we will be working on including your feedback to full extent in the paper itself. Please expect further improvements in the coming days. If there are any additional comments we are very interested to hear them.

---

> ### Comment · Reviewer_rQFP · 2023-11-23
> **Responses to authors' feedback**
>
> Thanks for the response. I appreciate the effort and the intuitive explanation makes sense to me. However, I do not completely buy your justification regarding the best-response dynamics, as what I was asking is better-response dynamics, which is a much broader class of dynamics compared to best-response. And also I do not completely agree with response 4, as the argument "This is because there are no CE/CCE which are different from Nash Equilibrium" does not make sense to me: in congestion games there is no evidence that CE/CCE coincide with NE. In addition, I do not believe the experiment section is comprehensive enough to support the theoretical claim. I would like to raise my score to 5 as I see the potential of this idea, however, I believe the overall quality of the current version is not good enough for acceptance.

---

### Author Response · Authors · 2023-11-22
**Final Rebuttal + Revised Paper**

Dear reviewers,

Thank you again for all your insightful feedback. We have thoroughly gone through all your comments and questions. In additions to the answers that we provided shortly after your reviews were released we are now happy to share with you a fully revised version of our manuscript. We believe that our work has been significantly improved based on the points that you have raised and we hope you will share this perspective. What we perceive as the biggest improvements is:
- the completely reworked theoretical sections. Based on your comments we had a critical overhaul of the theorems which we hope are now more precise and most importantly more understandable. We have also linked the theorems to our results and explicitly state what their value is.
- giving intuitions and examples as to the workings of the algorithm that we propose
- straightening out the clarity of the nomenclature
- highlighted another application setting for our algorithm - a non-augmented Braess network

With these (and other changes) we believe that the paper should now be more clear, more readable and our contribution more valuable to the community. We have highlighted all changes to our manuscript in \textcolor{blue}{blue} so that it is easy to see what has been changed. We hope that this work reflects the points that you have raised to your satisfaction. We would greatly appreciate your further engagement! Please let us say that we have been very happy to see you engage with our work in a meaningful and in-depth way and providing us with much needed critical perspective.

Kind Regards,
The Authors

p.s. Please find below a summary of all of our changes to be found in the Document:

---

> ### Author Response · Authors · 2023-11-22
> **Summary of All Changes to the Revised Paper**
>
> ## Changes and Edits
>
> - Section 1
>
>   - Section 1.1
>     - add comment motivating $Q$-learning given best-response dynamics converge to Nash Equilibrium
>     - add comment on improvements expected with recommendations given rational assumptions, or best-response assumptions
>   - Section 1.2
>     - add explanation of $\epsilon$-greedy
>     - add clarification of stateless MDP
>     - moved social welfare different from Nash Equilibrium comment
>     - add citation for Figure 2 results which should be used to get intuition about why $Q$-learners implicitly coordinate and do not converge to Nash in the Braess Paradox
>   - Section 1.3
>     - add references to mentioned work by reviewers on game theoretic models with strategic content providers
>
> - Section 2
>
>   - new word $Q$-users, to replace $Q$-learners for sake of clarity when referring to the users which play the congestion game
>   - *removed* for sake of clarity: ", an additional piece of information as the $Q$-users repeatedly play a congestion game"
>   - add small paragraph explaining the structure of following sections
>   - Section 2.1
>     - fix issue with KL divergence. Reward is now defined in terms of a distribution $P(a)$ over actions, rather than a distribution over action profiles.
>     - $\pi_{RS}^*$ is slightly adjusted to reflect dependence on $\pi_{RS}$ AND $\pi$, the policies of the users.
>   - Section 2.2
>     - reformatted to address only scaling related discussion
>   - proofs moved to section 2.3
>
> - Section 2.3 (NEW)
>
>   - Section 2.3.1 - Manipulating by increasing the recommendation space $Z$
>
>     - we have rephrased all of our proofs to be mathematical
>     - we have introduced a new definition of a "Reachable set", the set of actions that recommendations can induce in a q learner
>     - we have defined a new function $Ext$ which we use to extend $q$-tables as the size of the recommendation space increases
>     - new theorem "Increasing Reachability", which fixes our previous theorem 2
>     - new assumption "Full Support", which fixes our previous assumption 1
>
>     - new theorem "Global Reachability", which fixes our previous theorem 1
>
>     - these proofs justify a relationship between increasing recommendation spaces and the ability to manipulate the learning dynamics
>
>   - Section 2.3.2 - Manipulating By Optimizing Belief Updates
>
>     - in this section we explain how we can optimize the social welfare by manipulating the belief updates of the $Q$-learners
>     - previous pseudocode and subroutines have been moved to the appendix
>     - This section provides intuition about our design of the algorithm to justify why we do two main things:
>       - Positive Reinforcement: maximize the updates for socially beneficial actions
>       - Negative Reinforcement: minimize the updates for socially sub-optimal actions
>
>   - Section 2.3.3 - Constant recommendations change nothing
>
>     - In this section we explain why a constant recommendation is equivalent to no actions from the recommeder system through the use of an example
>
> - Section 3: Results
>
>   - No Changes
>
> - Section 4: Discussion and Conclusion
>
>   - add mention of future work on controllability of multi-agent systems
>   - add future work for larger congestion networks citing Braess-like networks in larger graphs
>   - add use for our recommender system approach to optimize social welfare while accounting for individual level metrics of content recommenders like engagement.

---

### Meta-Review · Area_Chair_cz2k · 2023-12-10

**Metareview:**

The reviewers had an overall negative sentiment towards the paper, which stems from the paper not meeting the bar of either a good empirical paper or a good theoretical paper, even though both were attempted in the draft. On the empirical front, reviewers pointed out that only a toy example simulation was done, thereby not being convincing enough to demonstrate the merit of the approach. On the theoretical front, a detailed theoretical characterization in this field is lacking, such as convergence towards a social optimal. The writing also has substantial improvements. The authors did attempt to make revisions during the rebuttal time to the draft. However, no consensus has been reached on whether the "home run" issue has been fully addressed. I see good promise of the paper and would encourage the authors to submit the fully revised paper to ICML.

**Justification For Why Not Higher Score:**

N/A

**Justification For Why Not Lower Score:**

N/A

---

### Decision · Program_Chairs · 2024-01-16

Reject